# Effect of Mechanical Loading of Senescent Myoblasts on Their Myogenic Lineage Progression and Survival

**DOI:** 10.3390/cells11243979

**Published:** 2022-12-09

**Authors:** Athanasios Moustogiannis, Anastassios Philippou, Evangelos Zevolis, Orjona S. Taso, Antonios Giannopoulos, Antonios Chatzigeorgiou, Michael Koutsilieris

**Affiliations:** 1Department of Physiology, Medical School, National and Kapodistrian University of Athens, 75 Micras Asias, 115 27 Athens, Greece; 2School of Biological Sciences, Deanery of Biomedical Sciences, Centre for Discovery Brain Sciences, Edinburgh EH8 9JZ, UK; 3Department of Surgical and Perioperative Sciences, Faculty of Medicine, Umeå University, 901 87 Umeå, Sweden

**Keywords:** mechanical loading, muscle cells, aging, cellular senescence

## Abstract

Background: During aging, muscle cell apoptosis increases and myogenesis gradually declines. The impaired myogenic and survival potential of the aged skeletal muscle can be ameliorated by its mechanical loading. However, the molecular responses of aged muscle cells to mechanical loading remain unclear. This study examined the effect of mechanical loading of aged, proliferating, and differentiated myoblasts on the gene expression and signaling responses associated with their myogenic lineage progression and survival. Methods: Control and aged C2C12 cells were cultured on elastic membranes and underwent passive stretching for 12 h at a low frequency (0.25 Hz) and different elongations, varying the strain on days 0 and 10 of myoblast differentiation. Activation of ERK1/2 and Akt, and the expression of focal adhesion kinase (FAK) and key myogenic regulatory factors (MRFs), MyoD and Myogenin, were determined by immunoblotting of the cell lysates derived from stretched and non-stretched myoblasts. Changes in the expression levels of the MRFs, muscle growth, atrophy, and pro-apoptotic factors in response to mechanical loading of the aged and control cells were quantified by real-time qRT-PCR. Results: Mechanical stretching applied on myoblasts resulted in the upregulation of FAK both in proliferating (day 0) and differentiated (day 10) cells, as well as in increased phosphorylation of ERK1/2 in both control and aged cells. Moreover, Akt activation and the expression of early differentiation factor MyoD increased significantly after stretching only in the control myoblasts, while the late differentiation factor Myogenin was upregulated in both the control and aged myoblasts. At the transcriptional level, mechanical loading of the proliferating myoblasts led to an increased expression of IGF-1 isoforms and MRFs, and to downregulation of muscle atrophy factors mainly in control cells, as well as in the upregulation of pro-apoptotic factors both in control and aged cells. In differentiated cells, mechanical loading resulted in an increased expression of the IGF-1Ea isoform and Myogenin, and in the downregulation of atrophy and pro-apoptotic factors in both the control and aged cells. Conclusions: This study revealed a diminished beneficial effect of mechanical loading on the myogenic and survival ability of the senescent muscle cells compared with the controls, with a low strain (2%) loading being most effective in upregulating myogenic/anabolic factors and downregulating atrophy and pro-apoptotic genes mainly in the aged myotubes.

## 1. Introduction

Aging is a complex process that involves a progressive decline in skeletal muscle mass and strength mediated by several cellular mechanisms, such as a negative muscle protein turnover, decreased production of anabolic factors, a disrupted regenerative capacity and myogenic lineage, and exacerbation of apoptosis [1,2,3,4]. A central characteristic of aging is cellular senescence [5], and a potential decrease in the number of muscle stem cells (MuSCs) might underlie the aging-related loss of muscle mass and impaired regenerative capacity [6,7].

Our knowledge regarding the molecular and cellular aspects of aging-induced muscle alterations can greatly be improved by utilizing in vitro models of muscle cell senescence [8,9]. Indeed, in a recent study, we revealed that the senescent myoblasts exhibited an increased expression of inflammatory, atrophy, and apoptotic genes, along with a reduction in the expression of myogenic regulatory factors (MRFs) and anabolic/metabolic factors, indicating a disrupted myogenic lineage and a reduced differentiation capacity of these senescent muscle cells [4].

Myogenesis is driven by different signal transduction pathways, which regulate the balance between muscle growth and atrophy [10], and in the context of myogenic differentiation, particularly MRFs, namely Myogenin, MyoD, myogenic factor 5 (Myf5), and myogenic regulatory factor 4 (MRF4), cooperatively regulate skeletal muscle growth and development [11,12]. Moreover, a central role in myogenesis has been documented for insulin-like growth factor 1 (IGF-1), while its isoforms may possess differential actions in muscle biology [13,14,15,16,17,18,19,20,21]. Two important signal transduction pathways, the mitogen-activated protein kinases (MAPK)/extracellular signal-regulated kinases (ERKs) and the AKT/mammalian target of rapamycin (mTOR)/P70S6K, are activated by IGF-1 [16,22], which induce MRFs expression and reduce proteolysis and the loss of skeletal muscle mass induced by the ubiquitin-proteasome system [23,24] and the Myostatin (MSTN)/Smad pathway [22], particularly in aging (sarcopenia) [25,26].

On the other hand, negative regulation of muscle growth and development is mediated by muscle atrophy and pro-apoptotic factors [27,28]. Specifically, studies have demonstrated increased Myostatin expression in skeletal muscle [29,30] and impaired IGF-1-induced myotube hypertrophy [31,32] with aging. Moreover, important enzymes in ubiquitin-mediated proteolysis and muscle atrophy have been identified, i.e., muscle ringer finger-1 (MuRF-1) and atrogin-1 (MaFbx) [33,34]. In addition, Fuca inhibits cell growth, the transcription factor Forkhead box O (FoxO is a fate decider in myogenic lineage [35], and, along with p53, it regulates diverse signaling pathways to control cell cycle and apoptosis [36], while they are negatively involved in muscle cell differentiation [37].

In our previous study, utilizing a myoblast aging model, we characterized the cellular senescence-induced responses of aged muscle cells during their myogenic differentiation [4]. Nevertheless, the mechanical loading-induced molecular mechanisms, which may regulate the myogenic lineage of the aged myoblasts, remain largely unknown. Skeletal muscle is able to adapt to mechanical loading, showing remarkable plasticity and altering its mass and contractile phenotype [38,39,40]. However, during aging, the processes of myogenic adaptations gradually decline and muscle cell apoptosis increases. Nonetheless, the mechanical loading of aged muscle has been shown to ameliorate its impaired myogenic potential, and in the last few years, in vitro cell loading models have been utilized to reveal the mechanisms of muscle adaption to mechanical loading [41].

Specifically, it has been shown that mechanical cues modulate many fundamental aspects of muscle cell function, such as proliferation and differentiation [42,43,44,45,46]. Moreover, the maintenance of muscle tissue homeostasis is critically dependent on mechanotransduction, i.e., the sense of external mechanical stimuli by mechanosensitive muscle cells and their conversion into intracellular biochemical signals [47,48]. The extracellular mechanical forces are transmitted to the cell nucleus through the interplay between mechanosensors, their corresponding signaling pathways, and various second messengers, which can modulate transcriptional activities for crucial cellular functions such as growth and survival [49]. In particular, focal adhesion kinase (FAK) is a pivotal mechanosensitive protein that is firstly recruited in response to mechanical stimuli [50,51,52]. More specifically, in skeletal muscle cells, the role of FAK is pleiotropic, regulating myogenesis, costamere formation, muscle phenotype and hypertrophy, and glucose uptake, and there is possibly a cross-talk between FAK and the phosphatidylinosital-3 kinase (PI3K) pathway [53], as well as with ERK 1/2 [54], while the integrin-associated FAK could be an important component of the IGF-1/PI3K/Akt pathway [55].

However, mechanosensing of the aged muscle cells may become dysfunctional due to their intrinsic changes induced by alterations in the expression of transmembrane and/or cytoskeletal proteins [7,56], while, further, the molecular responses of aged muscle cells to mechanical loading remain unclear. Thus, the purpose of this study was to examine the impact of the mechanical loading of aged myoblasts on the signaling and gene expression responses associated with their myogenic lineage progression and survival, ultimately aiming at contributing to a mechanistic model for the mechanotransduction alterations induced in aging muscle by myoblast senescence.

## 2. Materials and Methods

### 2.1. C2C12 Cell Culture

The mouse myoblasts C2C12 cell line was obtained from the American Type Culture Collection (Manassas, VA, USA) and was cultured as previously described [57]. Briefly, the cells were seeded onto six-well flexible-bottomed culture plates coated with Collagen I (Flex I Culture Plates Collagen I; Flexcell International, Hillborough, NC, USA), and were grown or differentiated before undergoing mechanical stretching, as described elsewhere [4,44].

### 2.2. C2C12 Myoblast Aging

To examine the responses of the aged skeletal C2C12 myoblasts to mechanical loading, a previously characterized model of muscle cell senescence was utilized [4]. Ultimately, 80 multiple doublings of myoblasts were completed over a period of 60 days to create the aged C2C12 cells.

### 2.3. C2C12 Cells Mechanical Loading

Aged and control C2C12 mononuclear cells and myotubes underwent passive cyclic stretching using the FX-5000 strain unit (Flexcell International, Hillborough, NC, USA), as previously described in detail [44]. Briefly, myoblasts and myotubes were subjected to two different stretching protocols: (a) low strain (2% elongation at a frequency 0.25 Hz for 12 h) and (b) high strain (15% elongation at a frequency of 0.25Hz for 12 h). Control and aged-myoblasts and myotubes without being stretched were used as the control condition.

### 2.4. Cell lysis, RNA Extraction, Reverse Transcription, and Real-Time PCR

C2C12 mononuclear cells and myotubes were lysed using NucleoZOL (Mecherey-Nagel, Duren, Germany) and cell extracts were collected as described previously [4,44]. The expression responses of the genes of interest were examined by reverse transcription and semi-quantitative real-time PCR using total RNA isolated 12 h after each stretching protocol, as described elsewhere [4,44]. The primer set sequences used for the specific detection of IGF-1 isoforms, MRFs, atrophy, and pro-apoptotic factors are presented in Table 1.

### 2.5. Protein Extraction and Immunoblotting Analysis

Total proteins were extracted from C2C12 cells and equal amounts of protein extracts (50 µg) were subjected to vertical electrophoresis (SDS-PAGE), then transferred to a polyvinylidene fluoride (PVDF) membrane (Bio-Rad, Hercules, CA, USA) and blots were incubated with the primary antibodies for the immunodetection of MyoD (1:1000 sc-377460; Santa Cruz, Dallas, TX, USA), Myogenin (1:1000, ab1835, Abcam, Cambridge, UK), p-Akt (1:2000, #9271; Cell Signaling, Danvers, MA, USA), p-Erk1/2 (1:2000, #9101; Cell Signaling), and FAK (1:500, #44-626G; Thermo Scientific, Waltham, MA, USA). The secondary antibodies used were anti-rabbit IgG (goat anti-rabbit, 1:2000; sc-2004) or anti-mouse IgG (goat anti-mouse, 1:2000; sc-2005). Glyceraldehyde 3-phosphate dehydrogenase (GAPDH) (1:2000, sc-47724) was used as a loading control. ImageJ software was used to semi-quantify the band intensity. A detailed description of all the above procedures is given elsewhere [4,20,44,58,59].

### 2.6. Statistics

The data are presented as mean ± standard error of the mean (SE). One-way ANOVA or Student’s *t*-test was used for statistics utilizing GraphPad Prism 5. Dunn’s multiple comparison was used as post hoc test. The level of statistical significance was set at *p* < 0.05.

## 3. Results

### 3.1. Effect of Mechanical Loading on Mechanosensitive and Signaling Proteins in C2C12 Myoblasts

To examine the activation of the mechanosensitive pathways by mechanical stretching in proliferating aged and control myoblasts, we investigated the effects of the two loading protocols on the expression of the mechanosensitive protein FAK and the phosphorylation of signaling proteins ERK1/2, which are also potential downstream effectors of FAK. Interestingly, differential responses were found in the aged myoblasts compared with the controls; a significant upregulation of FAK was induced by the low strain (2%) protocol in the control myoblasts and by the high strain (15%) in the aged cells (Figure 1A,C). Moreover, the activation of ERK1/2 in the aged myoblasts was induced only by the 2% stretching protocol (Figure 1B), while in the control cells both loading protocols resulted in increased phosphorylation of these signaling proteins (Figure 1B,D).

### 3.2. Effect of Mechanical Loading on MRFs in C2C12 Myoblasts

The effects of mechanical loading on the myogenic capacity of the proliferating aged and control myoblasts were examined by the expression levels of the early myogenic differentiation factors Myf5 and MyoD in the C2C12 cells. We found that in the aged myoblasts, only the low strain (2%) stretching protocol induced a significant increase in the expression of Myf5, in contrast with the control cells, in which both 2% and 15% elongation protocols resulted in significant upregulation of Myf5 (Figure 2A). Similarly, both loading protocols resulted in significant increases in the expression of the MyoD protein only in the control cells (Figure 2C,D), while at the transcription level, these increases were significantly higher compared with the responses of the aged cells to stretching (Figure 2B).

### 3.3. Effect of Mechanical Loading Protocols on Growth, Atrophy, and Pro-Apoptotic Factors in C2C12 Cells Myoblasts

We examined the effects of different mechanical stretching protocols on the expression of IGF-1 isoforms in proliferating aged and control myoblasts. Interestingly, both protocols caused significant upregulation of the IGF-1 isoforms only in the control and not in the aged myoblasts (Figure 3A,B). In addition, the effects of the two loading protocols on the transcriptional responses of muscle atrophy genes were examined in the aged and control myoblasts. It was found that significant decreases in the expression of all atrophy genes were induced only by the 2% strain protocol in the control, and not in the aged myoblasts compared with the non-stretched cells (Figure 3C–E). Moreover, apoptosis-related factors were examined in the proliferating myoblasts. Our findings showed that both mechanical loading protocols resulted in the upregulation of all the pro-apoptotic factors examined (Figure 3F–H), except for FoxO1, which was increased only by the 15% elongation protocol, and these increases were significantly higher compared with the responses to the 2% elongation protocol, both in the aged and the control cells (Figure 3F). Interestingly, mechanical loading caused the upregulation of FUCA only in the aged cells, regardless of the strain of loading (Figure 3G).

### 3.4. Effect of Mechanical Loading Protocols on Mechanosensitive and Signaling Proteins in C2C12 Myotubes

This study examined the activation of the mechanosensitive pathways by mechanical stretching, not only in the proliferating myoblasts, but also in differentiated (day 10 of differentiation) aged and control myotubes. Again, we investigated the potential effects of mechanical loading on FAK expression and the activation of its downstream effector Akt as a signaling protein in the PI3–kinase–Akt pathway, which is essential for muscle cell differentiation and survival. Similar responses to mechanical stretching were found in both the aged and the control myotubes, where a significant upregulation of FAK was induced by the low strain (2%) and the high (15%) strain loading protocol (Figure 4A,C). Interestingly, however, a significant activation (phosphorylation) of the signaling protein Akt was only observed in the control and not in the aged myotubes, regardless of the strain of mechanical loading (Figure 4B,D).

### 3.5. Effect of Mechanical Loading on Myogenic Regulatory Factors in C2C12 Myotubes

As MRFs largely regulate the myogenic program, we examined the effects of mechanical stretching on the myogenic capacity of aged and control C2C12 cells on day 10 of differentiation. The expression changes in early (MyoD) and late myogenic differentiation factors (Myogenin and MRF4) were investigated in myotubes in response to the different loading protocols. We revealed that the 2% strain protocol resulted in a significant decrease in the expression of MyoD only in the aged cells, and this downregulation was also significant compared with the responses of the aged cells to the 15% strain protocol (Figure 5A). At a protein level, the MyoD expression was significantly increased only by 2% mechanical loading in the control myotubes (Figure 5D). Interestingly, we found that the 2% elongation protocol induced significant increases in the mRNA and protein expression of the late differentiation factor Myogenin in both the aged and control myotubes (Figure 5B,E). Moreover, in the aged myotubes, the transcriptional upregulation of Myogenin by the 2% strain protocol was significantly different compared with the responses of the aged myotubes to the 15% strain protocol (Figure 5B). No significant changes were found in the protein expression of MyoD and Myogenin in both the aged and control cells in response to the 15% strain protocol (Figure 5F,G). Regarding the transcriptional responses of the other late myogenic factor, MRF4, to the different protocols, we revealed that it was significantly upregulated by the 15% strain protocol only in the control and not in the aged myotubes, while this upregulation was significantly different compared with the responses of the aged myotubes to the same protocol (Figure 5B,C).

### 3.6. Effect of Mechanical Loading οn Growth, Muscle Atrophy, and Pro-Apoptotic Factors in C2C12 Myotubes

IGF-1 has been documented to play a central role in myogenesis and thus the changes in the expression of IGF-1 isoforms as a result of different loading protocols were also examined in differentiated aged and control skeletal myotubes. We found that only the 2% stretching protocol induced significant increases in the expression of only the IGF-1Ea and not in the IGF-1Eb isoform in both the aged and control myotubes (Figure 6A,B). In addition, the expression responses of muscle atrophy genes and pro-apoptotic factors to the different stretching protocols were also examined in the differentiated myotubes. The significant downregulation of the muscle atrophy factors by the 2% strain protocol in both the aged and control myotubes and by the 15% strain protocol only in the control myotubes was revealed, while the decreases induced by the 2% strain were significant compared with the responses to the 15% elongation protocol in the aged myotubes (Figure 6C–E). Similarly, only the 2% elongation protocol resulted in significant decreases in the pro-apoptotic factors examined, in both the aged and control myotubes (Figure 6F–H), except for FoxO1, which was decreased only in the aged myotubes (Figure 6F). Again, in the aged myotubes, the decreases induced by the 2% strain protocol in the pro-apoptotic factors were significant compared with the responses to the 15% elongation protocol (Figure 6F–H).

## 4. Discussion

The present study investigated the effects of in vitro stretching of aged myoblasts and myotubes on the molecular responses associated with their myogenic progression and survival program. We utilized a characterized model of replicative myoblast senescence [4] and different mechanical loading protocols to ultimately reveal possible cell senescence-induced alterations in the mechanotransduction responses of aged muscle cells. Our main findings demonstrated a diminished beneficial effect of mechanical loading on the myogenic differentiation and survival potential of the senescent muscle cells compared with the controls. Moreover, a low strain/frequency loading protocol was revealed to be most effective in upregulating myogenic and anabolic genes and down-regulating atrophy and pro-apoptotic factors, mainly in the aged differentiated myotubes.

A variety of cellular responses, such as cell proliferation, survival, and apoptosis, can be modulated by mechanotransduction, while mechanical signals can particularly affect the myogenic potential of myoblasts to mature myotubes [44,60,61]. Indeed, one of the primary determinants of the skeletal muscle phenotype is the ability of muscle cells to respond to mechanical loading [48], which regulates their protein synthesis rate, while FAK is one of the first molecules that forms focal adhesions in response to mechanical stimuli, promoting cell survival, and the proliferation, motility, differentiation [62], and regulation of myogenesis [52]. In particular, the activation of FAK promotes the phosphorylation of Erk1/2 [63], which in turn stimulates the cellular proliferation [64], growth, and survival of muscle cells [65,66]. Moreover, FAK can interact with the PI3K/Akt pathway, which suppresses muscle atrophy and apoptosis and stimulates hypertrophy [54,67]. In general, the activation of the Ras/Raf/MEK/Erk1/2 signaling pathway in muscle cells has been documented to induce cell proliferation, while the activation of the PI3K/Akt pathway is involved in various cellular processes such as protein synthesis and cell survival [15,57,68,69,70,71,72].

In this study, we investigated the activation of Erk1/2 and Akt signaling proteins by mechanical stretching in both the aged proliferating myoblasts and the differentiated myotubes, as key components of the afore-mentioned mechanosensitive pathways and potential downstream effectors of the mechanosensitive protein FAK. It was revealed that the aged cells responded differentially compared with the control cells, as they exhibited only partial, or the absence of activation, of the ERK1/2 and Akt pathways, while, further, this activation appeared not to be associated with the overexpression of FAK, particularly in the differentiated myotubes. These findings indicate that in the aged muscle cells, these pathways may be less sensitive to mechanical signals regardless of the strain of mechanical loading. Indeed, it has been reported that aging reduces the sensation of muscle cells to adapt to mechanical signals [73], and previous studies have demonstrated that aging is associated with Akt dysfunction [74]. Moreover, a diminished myogenic/anabolic potential along with the induction of muscle atrophy and apoptotic factors were found to characterize the aged myoblasts [4].

Thus, in these senescent cells, we further investigated the effect of mechanical loading on their myogenic lineage and the anabolic potential again both in myoblasts and myotubes. MRFs function as the main transcription factors in myogenesis, activating muscle cell differentiation and regulating the expression of regulatory and structural muscle proteins, while MRFs have also been found to interact with growth factors, such as IGF-1 [16]. IGF-1 is largely involved in myogenesis and in the enhancement of muscle function in aging [14], while its isoforms have been reported to possess a differential role in aged muscle [4,75].

The present study showed that in the myoblasts, the early myogenic differentiation factors Myf5 and MyoD, as well as IGF-1 isoforms, were upregulated by mechanical loading protocols almost exclusively in the control and not in the aged cells. Interestingly, in the differentiated myotubes, the expression of MyoD was decreased in the aged cells and increased in the controls by the low strain (2%) loading protocol, while the same protocol resulted in the upregulation of the late myogenic factor Myogenin and the IGF-1Ea isoform in both the aged and control myotubes. The other late myogenic factor MRF4 was upregulated only in the control cells by the high strain (15%) loading protocol. Overall, these findings indicated differential responses of aged cells to mechanical loading compared with the controls, with the early and late differentiation MRFs, as well as the growth factor IGF-1, being sporadically affected in the aged myoblasts and myotubes only by the low strain loading protocol.

In parallel with the characterization of the anabolic potential of the aged cells in response to mechanical stretching, we investigated the expression responses of atrophy and pro-apoptotic factors to the different loading protocols in the aged myoblasts and myotubes. Muscle-specific atrophy factors, such as Myostatin and Atrogin-1, are considered negative regulators of the myogenic program [13,33,76,77], while many pro-apoptotic factors may also be involved in myogenesis [35]. Our findings indicated that, similarly to the occurrence of less myogenic/anabolic responses of the proliferating aged myoblasts to mechanical stimuli compared with the control cells, only the control and not the aged myoblasts exhibited a downregulation of muscle atrophy genes in response to the low strain loading protocol, while both stretching protocols resulted in pro-apoptotic factors being upregulated in the aged and control myoblasts. Interestingly, in the differentiated aged myotubes, the low strain, in contrast with the high strain loading protocol, resulted in the downregulation of muscle atrophy genes accompanied by anti-apoptotic responses, i.e., decreased expression of pro-apoptotic factors, similar to the responses of the control cells. Overall, considering our previous findings that the myogenic program of senescent myoblasts is accompanied by a less anabolic/more catabolic drive and increased activation of apoptotic responses compared to control cells, the present data suggest that a strain-specific mechanical loading of aged myotubes can reverse the anti-anabolic/apoptotic phenotype previously observed in the aged myoblasts during myogenesis. We recognize that using primary myogenic cells in the present study would widen the impact of our key innovative findings; however, this study was an important complementary continuance that completed our previous extensive characterization of the C2C12 myoblast senescence model in terms of their differentiation capacity and myogenic lineage [4]. Thus, expanding our experiments to primary cultures would firstly incur the need for an initial characterization of the “aged” primary cells, particularly in terms of their myogenic lineage throughout their differentiation program, as we did in our myoblast senescence model [4], before conducting the main mechanical loading experiments. Moreover, considering the need for ease of cell culture and accessibility due to the series of the subsequent mechanical loading experiments at different stages of myoblast differentiation, the primary cultures might be less efficiently, consistently, and “seamlessly” utilized. 

## 5. Conclusions

Muscle cells respond to mechanical signals in several ways so as to maintain their homeostasis and adapt to external loading, thus altering their mass and phenotype. During aging, the myogenic adaptations gradually decline and muscle cell apoptosis increases. This study demonstrated that aged myotubes retain their mechanosensitivity and respond differentially to strain-specific loading variations, exhibiting a myogenic/anabolic potential and anti-apoptotic responses to low strain mechanical stretching. These findings shed more light on the molecular responses of senescent muscle cells to mechanical signals, and thus might be a valuable resource for further characterizing the particular inputs of mechanotransduction pathways in the adaptations of aged muscle to mechanical loading. Understanding the molecular mechanisms of altered mechanotransduction in aging muscle will provide additional mechanistic insight and contribute to the development of strategies for the prevention and treatment of aging-associated skeletal muscle dysfunction.

## Figures and Tables

**Figure 1 cells-11-03979-f001:**
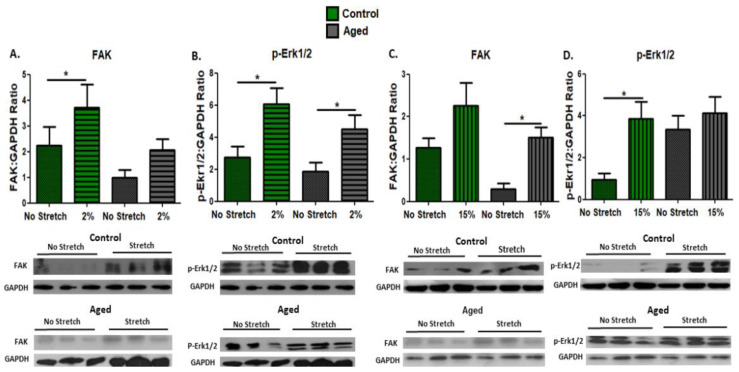
Effects of mechanical loading on the expression of the mechanosensitive protein FAK and the activation of the signaling proteins ERK1/2. Representative Western blots and immunoblotting quantification of (**A**) FAK, (**B**) p-Erk1/2 (2% strain), (**C**) FAK, and (**D**) p-Erk1/2 (15% strain) in the stretched, aged, and control myoblasts compared with the control condition (non-stretched cells). GAPDH on the same immunoblot was used as the internal standard (mean ± SE of three independent experiments performed in triplicate; significantly different: * *p* < 0.05).

**Figure 2 cells-11-03979-f002:**
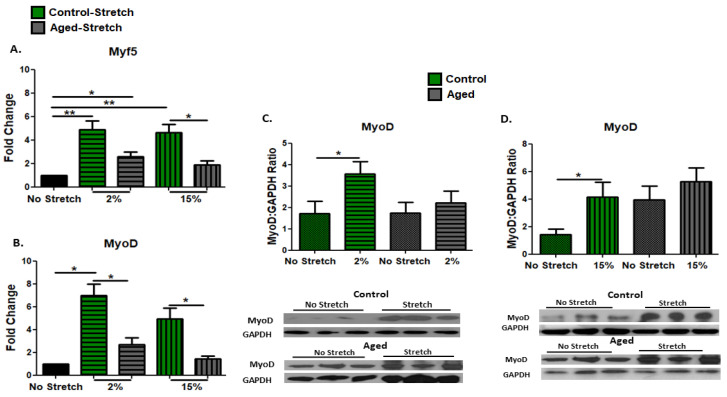
Effects of mechanical loading on the expression of myogenic regulatory factors. Analysis of (**A**) Myf5 and (**B**) MyoD mRNA expression in the stretched, aged, and control, myoblasts compared with the control condition (non-stretched myoblasts). The transcriptional responses of myogenic regulatory factors in the stretched cells were normalized to the corresponding GAPDH and are expressed as fold changes compared with the no stretch condition. Representative blots and immunoblotting quantification of MyoD (**C**,**D**) in the stretched and non-stretched, aged, and control myoblasts. GAPDH on the same immunoblot was used as the internal standard (mean ± SE of three independent experiments performed in triplicate; significantly different: * *p* < 0.05, ** *p* < 0.01).

**Figure 3 cells-11-03979-f003:**
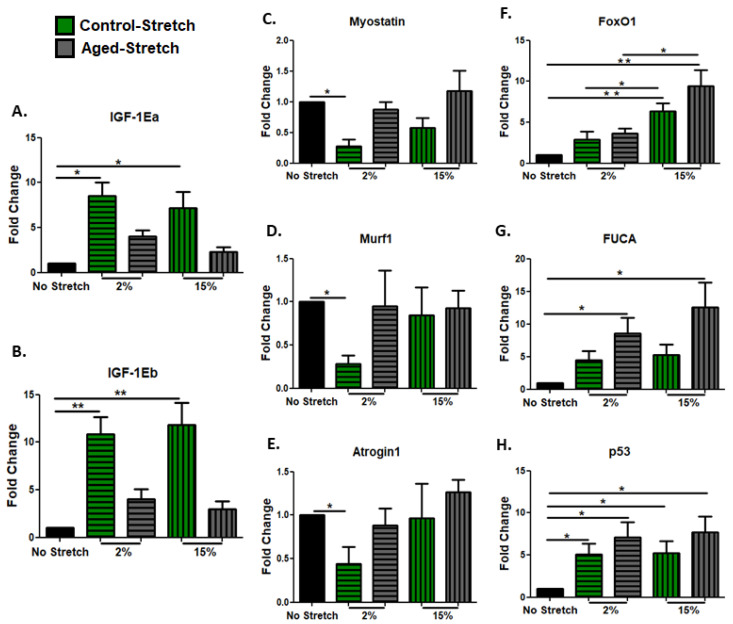
Mechanical loading-induced transcriptional changes in IGF-1 isoforms, muscle atrophy, and pro-apoptotic factors. Analysis of (**A**) IGF-1Ea, (**B**) IGF-1Eb, (**C**) Myostatin, (**D**) Murf1, (**E**) Atrogin1, (**F**) FoxO1, (**G**) FUCA, and (**H**) p53 mRNA expression in stretched and non-stretched aged and control C2C12 myoblasts. The transcriptional responses of the genes of interest in the stretched cells were normalized to the corresponding GAPDH mRNA and are expressed as fold changes compared with the no stretch condition (mean ± SE of three independent experiments performed in triplicate; significantly different: * *p* < 0.05, ** *p* < 0.01).

**Figure 4 cells-11-03979-f004:**
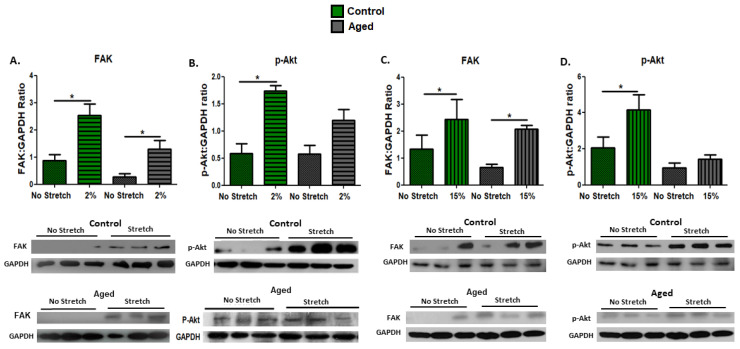
Effects of mechanical loading on the expression of the mechanosensitive protein FAK and the activation of the signaling protein Akt in differentiated myoblasts. Representative Western blots and immunoblotting quantification of (**A**) FAK, (**B**) p-Akt (2% strain), (**C**) FAK, and (**D**) p-Akt (15% strain) in stretched and non-stretched myotubes. GAPDH on the same immunoblot was used as the internal standard (mean ± SE of three independent experiments performed in triplicate; significantly different: * *p* < 0.05).

**Figure 5 cells-11-03979-f005:**
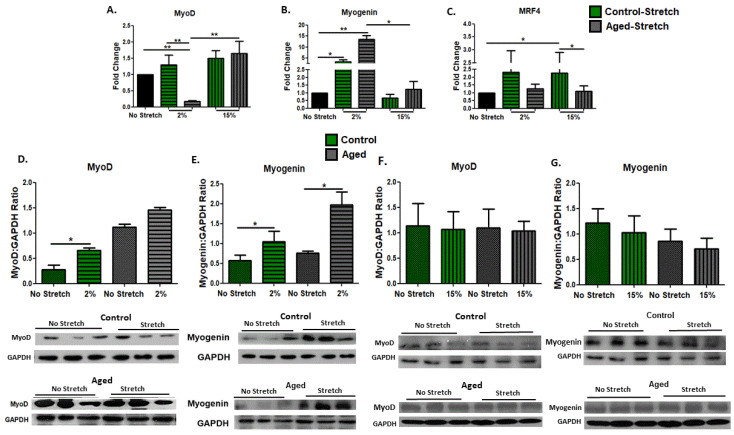
Mechanical loading-induced transcriptional changes of myogenic regulatory factors. Analysis of (**A**) MyoD, (**B**) Myogenin, and (**C**) MRF4 mRNA expression in stretched and non-stretched aged and control C2C12 myotubes. The transcriptional responses of the genes of interest in the stretched cells were normalized to the corresponding GAPDH and are expressed as fold changes compared with the no stretch condition. Representative blots and immunoblotting quantification of (**D**,**F**) MyoD and (**E**,**G**) Myogenin in stretched and non-stretched aged and control myotubes. GAPDH on the same immunoblot was used as the internal standard (mean ± SE of three independent experiments performed in triplicate; significantly different: * *p* < 0.05, ** *p* < 0.01).

**Figure 6 cells-11-03979-f006:**
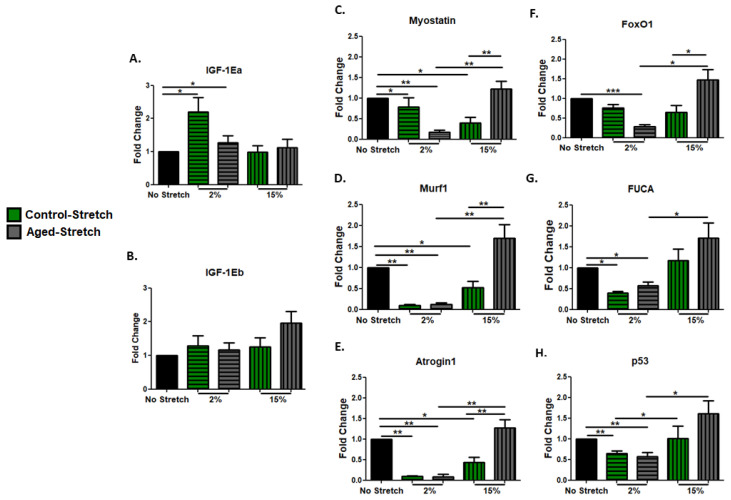
Mechanical loading-induced transcriptional changes of IGF-1 isoforms, muscle atrophy, and pro-apoptotic factors. Analysis of (**A**) IGF-1Ea, (**B**) IGF-1Eb, (**C**) Myostatin, (**D**) Murf1, (**E**) Atrogin1, (**F**) FoxO1, (**G**) FUCA, and (**H**) p53 mRNA expression in stretched and non-stretched aged and control C2C12 myotubes. The transcriptional responses of the genes of interest in the stretched myotubes were normalized to the corresponding GAPDH and are expressed as fold changes compared with the no stretch condition (mean ± SE of three independent experiments performed in triplicate; significantly different: * *p* < 0.05, ** *p* < 0.01, *** *p* < 0.001).

**Table 1 cells-11-03979-t001:** The sequence of the specific sets of primers used for RT-PCR analyses.

*Target Gene*	5′-3′ forward Primer Sequence	5′-3′ Reverse Primer Sequence	Product Length
*GAPDH*	*CAA CTC CCT CAA GAT TGT CAG CAA*	*GGC ATG GAC TGT GGT CAT GA*	*118*
*Myf5*	*CTA TTA CAG CCT GCC GGG AC*	*CTC GGA TGG CTC TGT AGA CG*	*232*
*MyoD*	*TGC TCC TTT GAG ACA GCA GA*	*AGT AGG GAA GTG TGC GTG CT*	*141*
*Myogenin*	*AGG AGA GAA AGA TGG AGT CCA GAG*	*TAA CAA AAG AAG TCA CCC CAA GAG*	*430*
*MRF4*	*AGG GCT CTC CTT TGT ATC CAG*	*TGG AAG AAA GGC GCT GAA GA*	*579*
*IGF-1Ea*	*GTG GAC GCT CTT CAG TTC GT*	*GCT TCC TTT TCT TGT GTG TCG ATA G*	*262*
*IGF-1Eb*	*GTC CCC AGC ACA CAT CGC G*	*TCT TTT GTG CAA AAT AAG GCG TA*	*259*
*FUCA*	*TTT GGT CGG TGA GTT GGG AG*	*CCA TTC CAA GAG CGA GTG GT*	*76*
*FoxO1*	*AGT GGA TGG TGA AGA GCG TG*	*GAA GGG ACA GAT TGT GGC GA*	*96*
*p53*	*GAG AGA CCG CCG TAC AGA AG*	*AGC AGT TTG GGC TTT CCT CC*	*317*
*Myostatin*	*CTG TAA CCT TCC CAG GAC CA*	*GCA GTC AAG CCC AAA GTC TC*	*104*
*MuRF1*	*AGG GCT CCC CAC CAC CTG TGT*	*TGC CCT CTC TAG GCC ACC G*	*310*
*Atrogin1* *(MAFbx)*	*AAC AAG GAG GTA TAC AGT AAG G*	*AAT TGT TCA TGA AGT TCT TTT G*	*322*

## Data Availability

Not applicable.

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
