# Peer review of "Effect of Mechanical Loading of Senescent Myoblasts on Their Myogenic Lineage Progression and Survival"

_cells, 2022, doi:10.3390/cells11243979_

Round 1

Reviewer 1 Report

The manuscript by Moustogiannis, et al, describes the effect on mechanical stretching on myogenic signaling pathways in young and aged myoblasts. Overall, I think that this study has been carried out well. However, there appear to be many similar studies reported for muscle cell stretching (recently reviewed in Stem Cells Dev. 2020 Mar 15;29(6):336-352), some of which investigate FAK and IGF. Therefore, to publish in the journal, Cells, with an impact factor above 7, I feel that further analysis is required to increase the impact of the results, which are detailed as follows:

Major point:

1) The analysis is carried out using a single murine myoblasts cell line, C2C12. To increase the impact of the results, I recommend repeating the major experiments using primary myogenic cells. These could be obtained as primary cultures from the skeletal muscle of young and aged mice. Alternatively, myogenic cells could be obtained from the young mice and artificially 'aged' using treatments such as ceramide ((Cells 2020, 9, 1385; doi:10.3390/cells9061385)). Another alternative would be to utilise human primary myoblasts, which are commercially available. With this additional data from primary myoblasts, I believe this manuscript may become suitable for publication in Cells.

Minor point:

1) On the title page, there appears to be no author for affiliation number 2.

Reviewer 2 Report

In this preclinical study, the authors examined the effects of in vitro stretching of aged myoblasts and myotubes on signaling and gene expression responses associated with progression of their myogenic and survival program, showing a decreased beneficial effect of mechanical load on myogenic capacity. and survival of senescent muscle cells compared to controls.

The methodology is very interesting and adequate to address the objective of the study. It is clearly exposed, from the description of stretching and C2C12 culture to the design of the groups and the different techniques used. The results are clearly displayed both in the text and in the figures.

Overall it is a good job, well done and well presented. I have only one criticism:

Why did the authors choose gapdh as housekeeping for their RT-PCR and WB analysis? For RT-PCR, did you perform a housekeeping assay using a panel of genes in order to choose and use GAPDH as housekeeping in this cell model? While for WB tecnique, I often see very large bands that are difficult to analyze. Can the authors attach the full membrane of the WB?

Round 2

Reviewer 1 Report

I believe that the authors have justified their use of C2C12 cells for this study, rather than primary cells. I recommend that this explanation should be included as an extra paragraph in the Discussion section of the manuscript, because some readers may not be aware of the reason for focusing on C2C12 alone for these studies of mechanical loading.

Author Response

Comments and Suggestions for Authors

I believe that the authors have justified their use of C2C12 cells for this study, rather than primary cells. I recommend that this explanation should be included as an extra paragraph in the Discussion section of the manuscript, because some readers may not be aware of the reason for focusing on C2C12 alone for these studies of mechanical loading.

Reply to Reviewer 1

We really appreciate the reviewer’s comments and suggestions that helped us to improve our manuscript. According to his/her suggestions the following text was added to the Discussion section:

“ We recognize that using primary myogenic cells in the present study would widen the impact of our key innovative findings; however, this study was an important complementary continuance that completed our previous extensive characterization of C2C12 myoblast senescence model in terms of their differentiation capacity and myogenic lineage [4]. Thus, expanding our experiments to primary cultures would firstly incur the need of an initial characterization of the “aged” primary cells, particularly in terms of their myogenic lineage throughout their differentiation program, as we did in our myoblast senescence model [4], before conducting the main mechanical loading experiments. Moreover, considering the need for ease of cell culture and accessibility due to the series of the subsequent mechanical loading experiments at different stage of myoblast differentiation, the primary cultures might be less efficiently, consistently, and “seamlessly” utilized. ”